# Mitigating Text Degeneration via Token-Level Guidance For pruned Large Language Models

## Abstract

Large language models (LLMs) suffer from substantial memory and inference costs, and pruning has emerged as a widely adopted strategy for compression. However, while pruning effectively reduces model size and latency, it often exacerbates undesirable side effects such as text degeneration, particularly repetition, even when perplexity remains largely intact. We observe that standard post-pruning fine-tuning is insufficient to suppress repetition, motivating the need for more targeted approaches. To address this issue, we propose two token-level guidance methods: **FOCUS** and **RePAIR**. FOCUS adjusts token probabilities through token probability weighted distillation, focusing on high-confidence regions to better align the student with the teacher while reducing the likelihood of undesirable tokens. In contrast, RePAIR employs contrastive training with negative and positive samples to explicitly encourage the generation of alternative tokens. Experiments on open-ended and instruction-based generation tasks demonstrate that our methods substantially reduce repetition and improve generation diversity, while causing only minimal impact on perplexity. Furthermore, our methods are compatible with other training strategies and consistently enhance their performance. Code will be available soon.

## 1 Introduction

Large language models (LLMs) have become foundational in a broad spectrum of applications, ranging from dialogue systems and code assistants to knowledge-intensive question-answering and content generation (Minaee et al., 2025; Jiang et al., 2024; Yue, 2025; Naveed et al., 2024). In practical deployments, however, the execution of large models is far from straightforward: Inference latency escalates dramatically under constrained serving budgets, and memory requirements often exceed the limits of commodity hardware (Chitty-Venkata et al., 2024; Pope et al., 2022). To manage these constraints, prior researches have developed pruning strategies, ranging from depth pruning, which eliminates entire transformer blocks, to width pruning, which removes internal submodules such as attention heads or MLP channels (Ma et al., 2023; Kim et al., 2024; Frantar & Alistarh, 2023). Because pruning inevitably discards parameters that encode useful behaviors, practitioners typically perform a post-pruning finetuning stage to restore degraded capabilities. For example, the techniques recover knowledge in pruning settings with a relatively small dataset, e.g., the Alpaca dataset with its instruction–response pairs (Taori et al., 2023).

Most prior work on pruning has focused on knowledge preservation (Li et al., 2024; Park et al., 2024). Standard evaluations assess whether a pruned model preserves perplexity, zero-shot accuracy, and downstream task performance relative to its unpruned counterpart. However, there is growing empirical evidence that pruning can also introduce undesirable side effects, *even when* principal metrics such as perplexity or accuracy appear to be intact (Liebenwein et al., 2021; Jordao & Pedrini, 2021; Jaiswal et al., 2024). A primary issue is text degeneration, where the model **repeatedly generates** the same words or phrases, for example: "<prefix> ... for the deceased. The cemetery is designed to be a peaceful place. The cemetery is designed to be a peaceful place. The cemetery is designed to be a peaceful place. ...". To demonstrate this, we generate 200 tokens from the WikiText-103 dataset using a 50-token prefix with both the unpruned LLaMA model and the pruned model after fine-tuning. A sentence is classified as repetitive if repeated segments account for the majority

of the generated text, as further discussed in Section 3. As shown in Table 1, the degeneration phenomenon becomes more severe after pruning in both cases. This observation indicates that, although simple fine-tuning can recover knowledge to some extent, it remains essential to mitigate the side effects that arise during text generation.

Previous studies have noted that text degeneration occurs when previously generated tokens increase the likelihood of the model producing the same tokens again (Holtzman et al., 2020; Welleck et al., 2019; Xu et al., 2022). To address this, Their approaches lower the probabilities of previously generated tokens while increasing those of tokens that have not appeared. Although this strategy is effective in reducing repetition, it does not provide guidance on which tokens should be generated to produce a coherent continuation, resulting degraded perplexity. Ideally, a method should not only suppress repetition but also encourage the generation of plausible tokens that contribute to better continuation.

Motivated by this, we propose two token-level guidance methods: **FOCUS** and **RePAIR**. Specifically, during the fine-tuning stage after pruning, **FOCUS** adjusts token probabilities through token probability weighted distillation, effectively reducing the likelihood of positive feedback from previous token. In contrast, **Re-PAIR** exploits pairs of negative samples (degenerated continuations) and positive samples (regenerations from the onset of degeneration), and applies a margin loss to explicitly encourage the generation of alternative tokens. To val-

| Condition | Sampling | Greedy |
|---|---|---|
| Unpruned | 5.9% | 26.6% |
| Width pruned | 12.4% | 63.1% |
| Depth pruned | 15.4% | 63.7% |

Table 1: Comparison of repetition rates between unpruned and pruned models. The pruned model is finetuned on the Alpaca dataset. For decoding, we employ top-$p$ sampling with $p = 0.9$.

idate our approach, we first prune the model and fine-tune it with LoRA in combination with our proposed methods. We then evaluate performance on two tasks: open-ended generation and instruction-based generation. The results demonstrate that our methods effectively reduce text degeneration with only a slight increase in perplexity. For example,in the open-ended generation task, our method achieves the highest MAUVE score and a comparable level of repetition on WikiText-103, indicating that the generated text more closely aligns with real text. Also, in the directed generation task, our method achieves the highest $EAD_1$ and BERTScore, reflecting improved diversity and closer alignment with reference texts. Moreover, we show that other training-based methods, when combined with FOCUS, can also benefit from enhanced generation quality.

We summarize our contribution as follows:

- Through our analysis, we show that text degeneration can be mitigated by providing plausible alternative tokens. Furthermore, we observe that naive knowledge distillation fails to fully transfer the teacher's knowledge due to the capacity gap between teacher and student models, often leading the student to assign higher probabilities to tokens that the teacher intends to suppress.

- Motivated by our analysis, we propose two simple yet effective methods, FOCUS and RePAIR. This approach provides token-level guidance by only focusing on regions where the teacher is confident, while simultaneously encouraging the generation of plausible alternative tokens.

- Our experiments demonstrate that the proposed methods on open-ended and instruction-based generation. Our methods improve generation quality in metrics such as unique $n$-gram, MAUVE and $EAD_1$, while causing only minimal impact on perplexity.

## 2 RELATED WORK

### 2.1 LLM PRUNING & DISTILLATION

As large language models (LLMs) scale in capacity, their memory footprint and inference latency have emerged as critical bottlenecks, particularly in resource-constrained or real-time deployment scenarios (Yao et al., 2025b; Tian et al., 2025). To mitigate these issues, model pruning has been extensively investigated as an effective strategy to reduce model size and computational overhead. However, pruning inevitably introduces knowledge loss and results in performance degradation (Kim et al., 2024; Ma et al., 2023). To address this, knowledge distillation (KD) is commonly

employed in the post-pruning fine-tuning stage to transfer knowledge from the teacher to the pruned student model (Xu et al., 2024; Gu et al., 2024), offering the advantage of leveraging soft probabilities to provide richer supervision than one-hot labels. Nevertheless, KD does not always yield consistent benefits (Ma et al., 2021; Zhang et al., 2025). For example, naive KD may transfer incorrect answers from the teacher and they demonstrate that student models trained with appropriate corrections of teacher logits can even outperform the teacher itself in classification tasks (Zhang et al., 2024). Also, several studies have also noted that naive knowledge distillation may fail to effectively train the student model, either due to the capacity gap between teacher and student or biases inherent in the teacher model (Zhong et al., 2024; Shum et al., 2024). These observations highlight the need for strategies that selectively extract and transfer only useful information from the teacher to improve training effectiveness.

## 2.2 Mitigation of Text Degeneration

Degeneration in text generation has been addressed through both decoding-time and training-time strategies. On the decoding side, deterministic methods such as greedy search and beam search are widely used, but they often suffer from limited output diversity and frequent degeneration. Top-$k$ sampling, which restricts decoding to the $k$ most probable tokens at each step to filter out low-probability candidates while maintaining diversity, was proposed in earlier work (Fan et al., 2018). A more flexible alternative, top-$p$ (nucleus) sampling, dynamically selects the smallest set of tokens whose cumulative probability exceeds a threshold $p$, as introduced by (Holtzman et al., 2020). By adapting to the sharpness of the probability distribution, top-$p$ sampling produces more natural and diverse outputs.

Complementary to decoding-based methods, training-based approaches modify the learning objective to directly discourage repetition. Prior work has observed that the probability of previously generated tokens tends to be amplified, increasing the likelihood of their recurrence in subsequent positions. To address this issue, Unlikelihood Training adds an auxiliary loss that penalizes repeated tokens by reducing their probabilities (Welleck et al., 2019). ScaleGrad instead adjusts token-level gradients during training, encouraging the generation of novel tokens while suppressing those already produced (Lin et al., 2021). However, the unlikelihood-based approach often deteriorates perplexity. More recently, DITTO constructs synthetic sentence-level repetition data and trains the model by explicitly reducing the probability of later repeated tokens relative to earlier occurrences by a factor of $n$ (Xu et al., 2022).

## 3 Token-level guidance

In this section, we first demonstrate that degeneration can be disrupted by providing token-level guidance. We further show that naive KD fails to transfer the teacher distribution, leading the student model to learn a flattened distribution that allocates non-negligible probability mass to tokens the teacher would not intend to sample. Before presenting these results, we define **Coverage** and introduce **CREP**, a **C**overage-based **REP**etition metric designed to assess whether a generated text is degenerated. Coverage and CREP are formally defined as follows:

$$\text{Coverage}(r, s_{1:T}) = \frac{1}{T} \sum_{j=1}^{T-N+1} \mathbf{1}[\, s_{j:j+N-1} \approx r \,] \cdot N$$

$$\text{CREP}(D) = 100 \times \frac{1}{|D|} \sum_{s \in D} \mathbf{1}[\, \text{Coverage}(r, s) \geq \tau \,],$$

This metric computes the fraction of tokens in a sentence $s_{1:T}$ that are covered by occurrences of an n-gram $r$. A higher coverage value indicates that the $n$-gram occupies a larger portion of the text, which can serve as evidence of degeneration such as repetition. **CREP**, in turn, measures the proportion of sentences in the dataset $D$ whose Coverage exceeds a threshold $\tau$, thereby reflecting the overall rate of degenerated sentences.

## 3.1 SIMPLE TOKEN GUIDANCE EXPERIMENT

To demonstrate that degeneration can be allevi- ated by avoiding entry into repetition loops, we first generate 1,000 sentences under the same settings described in Section 1. We then fil- ter degenerated sentences using the **CREP** met- ric and identify the onset of the first repetition loop. Next, we replace the first two tokens with alternatives chosen from the top-2 probability candidates and regenerate an additional 200 to-

| Condition | Before | After |
|-----------|--------|-------|
| Sampling | 6.4% | 0.7% |
| Greedy | 26.6% | 10.8% |

Table 2: Unpruned model degeneration rates be- fore and after correction.

kens, re-evaluating the repetition metric. As shown in Table 2, once the first two tokens are substi- tuted with other plausible candidates, the repetition ratio decreases from 6.4% to 0.7%. This result indicates that degeneration can be effectively alleviated through token-level guidance. However, in practical generation scenarios, anticipating in advance which tokens will trigger degeneration is non-trivial. This makes real-time guidance infeasible, thereby motivating the need for preemptive training-time strategies.

## 3.2 DISTILLATION PROCESS IN LLM

As discussed earlier, naive KD may inadvertently prop- agate negative effects. A well-known example is *mode averaging*, where the student spreads its probability mass too widely in order to cover the teacher distribution (Wu et al., 2024; Yao et al., 2025a; Ko et al., 2024). The KL divergence, which is commonly used in KD, is defined as $\mathcal{L}_{\text{KL}}(q\|p) = \sum_x q(x) \log \frac{q(x)}{p(x)}$, where $q$ denotes the teacher distribution and $p$ the student distribution. This formulation has two key characteristics: (i) it penalizes the student for underestimating regions where the teacher assigns probability mass, and (ii) it barely penalizes prob- ability placed on regions the teacher assigns zero. As il- lustrated in Figure 1, when training a simple Gaussian (student) against a more complex mixture of Gaussians (teacher) under KL minimization, the student often allo- cates probability mass to regions that the teacher does not

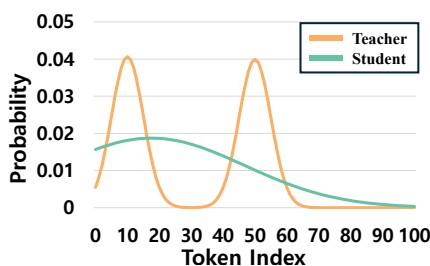

Figure 1: Illustration of mode averaging under KL minimization. The curve for the student corresponds to the numeri- cally computed optimum.

intend to support covering broad region. In the context of text generation, this implies that the stu- dent may raise the probability of tokens that the teacher considers unlikely, thereby increasing the chance of producing contextually inappropriate words and exacerbating degeneration. This obser- vation suggests that a student with lower capacity cannot effectively capture the full capacity of the teacher distribution. Instead, it is crucial to design a distillation scheme that emphasizes informative regions of the distribution on a token-level, rather than indiscriminately imitating the entire teacher distribution.

# 4 METHOD

In this section, we propose the token probability weighted distillation method, which follows the trend of useful teacher probability but mitigates degeneration. Next, we propose the pairwise margin loss, which directly guides the model to generate probable alternative tokens.

## 4.1 FOCUS: TOKEN PROBABILITY WEIGHTED KNOWLEDGE DISTILLATION

**FOCUS training objective.** Given a dataset $\mathcal{D} = \{s_i\}_{i=1}^N$, where each sequence is tokenized as $s_i = (s_{i,1}, \ldots, s_{i,T})$, let $z^T(s_{i,<t}) \in \mathbb{R}^V$ denote the teacher logits over the vocabulary $\mathcal{V}$ at position $t$. We define the student and teacher predictive distributions as

$$p_{i,t} \equiv p_\theta(\cdot \mid s_{i,<t}), \qquad q_{i,t} \equiv \text{softmax}\left(\frac{z^T(s_{i,<t})}{\tau}\right),$$

We introduce a token-wise weight

$$w_{i,t}(s_{i,t}) \;=\; \big(q_{i,t}(s_{i,t})\big)^{\beta} \;+\; \big(1 - q_{i,t}(s_{i,t})\big)^{\gamma}, \quad \beta, \gamma \geq 0,$$

which emphasizes tokens where the teacher exhibits high confidence. Thus, naive KD is reformulated as

$$\mathcal{L}_{\text{FOCUS}} = \frac{1}{NT} \sum_{i=1}^{N} \sum_{t=1}^{T} \left( \tau^2 \sum_{v \in \mathcal{V}} w_{i,t}(v)\, q_{i,t}(v) \, \log \frac{q_{i,t}(v)}{p_{i,t}(v)} \right).$$

**Gradient Analysis of FOCUS**    From the FOCUS objective

$$L_{\text{FOCUS}} = \sum_i w(q_i)\, q_i \log \frac{q_i}{p_i},$$

Applying the chain rule,

$$\frac{\partial L}{\partial a_k} = \sum_i \frac{\partial L}{\partial p_i} \frac{\partial p_i}{\partial a_k} = Z p_k - w_k q_k, \qquad \text{where } Z = \sum_j w_j q_j.$$

Defining the reweighted teacher distribution and gradient can be expressed as

$$\tilde{q}_k = \frac{w_k q_k}{Z}, \qquad \nabla_a L_{\text{FOCUS}} = Z\,(p - \tilde{q}).$$

Thus, FOCUS preserves the standard KD form while effectively replacing the teacher distribution with a reweighted version $\tilde{q}$. The detailed derivation is provided in the Appendix A.

## 4.2    RePAIR: **R**epetition-aware **PAIR**wise Alignment

As discussed in Section 3, token-level guidance can alleviate repetition, but it cannot be applied at runtime since detecting the onset of a repetition loop requires access to future tokens. To address this, We propose RePAIR, a repetition-aware pairwise alignment that provides token-level corrective signals exactly where degeneration begins, guiding the student toward non-repetitive and contextually coherent generation. An example is provided in Appendix B.

### 4.2.1    Pairwise data collection

Given a prefix length $k$, we generate model outputs $\hat{y}_i$ from inputs $s_{i,0:k}$. Using the Coverage metric described in Section 3, sequences with repetition above threshold are collected as negative samples $D_{\text{neg}}$. The index $r_i$ of the first repetition defines a shorter prefix $s_{i,0:(r_i-1)}$, from which we regenerate outputs to obtain positive samples $D_{\text{pos}}$. In practice, negative data are produced by the pruned model, while positive data are sampled from the unpruned model with top-$p$ decoding, yielding realistic pruned/unpruned comparisons. We set threshold as 0.3.

### 4.2.2    RePAIR training objective

We define a token-level pairwise margin loss to encourage the model to assign higher confidence.

**Trainig objective.**    Let $p_\theta(\cdot \mid \cdot)$ denote the model distribution. We compute token-level negative log-likelihood only for the continuation after the prefix $s_{i,0:(r_i-1)}^{\text{pre}}$.

$$\ell_i^{+} = -\frac{1}{T_i^{+} - r_i + 1} \sum_{t=r_i}^{T_i^{+}} \log p_\theta\big(s_{i,t}^{\text{pos}} \mid s_{i,0:(r_i-1)}^{\text{prefix}}, s_{i,<t}^{\text{pos}}\big),$$

$$\ell_i^{-} = -\frac{1}{T_i^{-} - r_i + 1} \sum_{t=r_i}^{T_i^{-}} \log p_\theta\big(s_{i,t}^{\text{neg}} \mid s_{i,0:(r_i-1)}^{\text{prefix}}, s_{i,<t}^{\text{neg}}\big).$$

We encourage the model to prefer the positive continuation over the negative one by a margin-ranking loss:

$$\mathcal{L} = \frac{1}{N} \sum_{i=1}^{N} \max\big(0, \; m + \ell_i^{+} - \ell_i^{-}\big). \tag{1}$$

### 4.2.3 Total loss function

The final objective can be expressed as

$$L_{\text{total}} = L_{\text{CE}} + \alpha_1 \cdot L_{\text{FOCUS}} + \alpha_2 \cdot L_{\text{RePAIR}}$$

Unless otherwise specified, $\alpha_1$ is fixed at $0.05$ and $\alpha_2$ at $1$ for all experiments. A detailed discussion on parameter selection is provided in Appendix E.

## 5 Experiments

### 5.1 Setup

**Training & Evaluation** For our experiments, we use the LLaMA family, model that is widely adopted in practice. We prune 25% of the model using LLMPruner, which reduces the width of the model, and subsequently fine-tune it on the Alpaca dataset, a standard benchmark for restoring knowledge after pruning (Taori et al., 2023). Fine-tuning is conducted with LoRA for two epochs. To further validate the generality of our method under more aggressive model pruning, we also evaluate models pruned at 35% and 45%. The corresponding results are provided in Appendix G.

We evaluate our approach on two tasks: open-ended generation and instruction-based generation. For open-ended generation, following prior works, we randomly sample 1,000 instances from the WikiText-103 dataset and generate 100 tokens conditioned on a 50-token prefix. For instruction-based generation, we utilize 1,000 prompts from the Self-Instruct dataset (Wang et al., 2023) to examine whether the pruned model can retain general knowledge while avoiding repetition. In both tasks, we use top-$p$ sampling with $p = 0.9$. Finally, we evaluate the generated outputs along two dimensions: task performance and text degeneration.

**Performance Metrics** After pruning and fine-tuning, the model should maintain fluency. We report performance using the following metrics:

- **Perplexity**: Perplexity (PPL) measures how well a language model predicts the next token, with lower values indicating better predictive performance.

- **BERTScore (BS):** BERTScore evaluates the semantic similarity between generated text and reference text by computing the cosine similarity of contextualized embeddings from a pretrained BERT model (Zhang et al., 2020). We report the F1 score.

- **Zero-shot Accuracy**: We evaluate zero-shot accuracy by testing the model on a diverse set of tasks without task-specific fine-tuning. This metric reflects the model's ability to apply its general knowledge and reasoning skills in unseen settings, and we report the mean accuracy across tasks. Further details on the tasks and evaluation settings can be found in the Appendix C.

- **MAUVE**: MAUVE measures the distributional similarity between generated and reference sentences in the embedding space. When the model produces low-quality outputs, such as hallucinations or repetitions, the distance between the two distributions increases, resulting in a lower score (Pillutla et al., 2021).

**Degeneration Metrics** To capture the degeneration issues of pruned models, we measure repetition tendencies. Specifically, we analyze:

- **Unique $n$-gram**: To evaluate the diversity in a sentence, Unique n-gram Rate is widely used and defined as $= 100 \times (1 - \frac{1}{N} \sum_{i=1}^{N} \frac{|\text{Unique } n\text{-gram}(\text{sentence}_i)|}{|\text{Total } n\text{-gram}(\text{sentence}_i)|})$. we set $n = 3, 4, 5, 6$ to capture both word-level and phrase-level diversity (Xu et al., 2022).

- **Expected 1-gram Diversity (EAD$_1$):** EAD$_1$ quantifies lexical diversity by comparing the number of unique tokens in the generated text to the expected count under random sampling. A higher value indicates greater diversity, and it is formally defined as $\frac{N}{V_{\text{eff}}\left(1 - \left(1 - \frac{1}{V_{\text{eff}}}\right)^C\right)}$, where $N$ is the number of unique tokens in the generated text, $C$ is the total number of generated tokens, and $V_{\text{eff}}$ is the effective vocabulary size (Liu et al., 2022).

- **CREP**: While the proportion of unique $n$-grams reflects sentence diversity, it does not directly capture whether a sentence is degenerated. To complement this, we use **CREP**, which measures

Table 3: Results of open-ended generation on the WikiText-103 dataset. Best per block in **bold**, second best underlined. CREP and Unique $n$-gram are reported as percentage values.

| Method | PPL (↓) | 0-shot (↑) | MAUVE (↑) | CREP (↓) | Unique $n$-gram | | | |
|---|---|---|---|---|---|---|---|---|
| | | | | | $n=3$ | $n=4$ | $n=5$ | $n=6$ |
| LLaMA-3.1-8B | | | | | | | | |
| KD | 21.69 | 60.39 | 0.61 | 7.3 | 13.28 | 9.99 | 8.05 | 6.79 |
| + UL | **21.67** | 61.15 | 0.61 | 5.37 | 11.79 | 8.68 | 6.85 | 5.70 |
| + SG | 21.69 | 60.88 | 0.66 | 7.8 | 12.87 | 9.69 | 7.85 | 6.66 |
| + DITTO | 22.07 | 60.62 | 0.63 | 5.27 | 11.38 | 8.14 | 6.22 | 5.01 |
| + RePAIR | 22.05 | **61.36** | **0.68** | **2.23** | 8.36 | 5.36 | 3.74 | 2.78 |
| FOCUS | **22.32** | 60.03 | 0.64 | 1.73 | 6.22 | 3.82 | 2.62 | 1.94 |
| + UL | 23.20 | 60.45 | 0.69 | 0.77 | 4.30 | 2.43 | 1.51 | 1.07 |
| + SG | 22.71 | 60.47 | 0.70 | 0.87 | 4.86 | 2.89 | 1.96 | 1.47 |
| + DITTO | 22.88 | 60.54 | 0.72 | **0.5** | 4.49 | 2.42 | 1.46 | 1.11 |
| + RePAIR | 23.12 | **60.86** | **0.73** | 0.57 | 3.68 | **1.92** | **1.14** | **0.75** |
| LLaMA-2-13B | | | | | | | | |
| KD | **13.90** | 64.10 | 0.81 | 1.13 | 6.84 | 4.02 | 2.49 | 1.67 |
| + UL | **13.90** | 64.28 | 0.79 | 1.50 | 6.73 | 4.02 | 2.58 | 1.78 |
| + SG | **13.90** | 64.15 | **0.82** | 1.20 | 6.87 | 4.08 | 2.59 | 1.78 |
| + DITTO | 14.19 | 63.85 | 0.74 | 1.30 | 7.14 | 4.24 | 2.70 | 1.84 |
| + RePAIR | 13.95 | **64.36** | 0.81 | **0.50** | 5.55 | 2.97 | 1.67 | **1.01** |
| FOCUS | **15.41** | 63.69 | 0.84 | 0.13 | 2.84 | 1.20 | **0.55** | 0.29 |
| + UL | 15.74 | 63.57 | 0.85 | 0.07 | 2.40 | 0.94 | 0.38 | 0.17 |
| + SG | 15.47 | 63.77 | 0.84 | 0.10 | **2.57** | **1.07** | 0.47 | **0.23** |
| + DITTO | 15.85 | 63.52 | 0.86 | 0.03 | 2.46 | 0.93 | 0.37 | 0.16 |
| + RePAIR | 15.50 | **63.91** | **0.87** | **0.00** | 2.24 | 0.82 | 0.32 | 0.13 |
| WikiText-103 | - | - | - | - | 2.62 | 1.10 | 0.52 | 0.24 |

the fraction of sentences where repeated $n$-grams dominate more than $30\%$ of the text. Specifically, for each sentence, we evaluate repetition across a range of $n$-gram sizes, varying $r$ from 4 to 16, and mark the sentence as degenerated if *any* of these $r$ values exceeds the threshold. This metric provides a clear indication of how many sentences are degenerated, offering an intuitive view of repetition severity. For detailed implementation, we leave a pseudo code in Appendix F.

**Comparison Methods** We compared our model with five training based methods: KD Finetune(KD) finetunes it with Knowledge distillation method in convention after pruning the model. UL-Token: penalizes tokens previously generated (Welleck et al., 2019). Although sentence-level unlikelihood can also be applied, it is often impractical for large-scale LLMs such as LLaMA due to computational constraints. ScaleGrad(SG) re-normalize the softmax distribution of tokens after scaling probability of tokens not generated (Lin et al., 2021). DITTO constructs sentence-repetition datasets by repeating the existing sentence and penalizes the probability of generating repetitive sequences (Xu et al., 2022). All methods are implemented following the settings of their original papers and leave a settings in Appendix D.

## 5.2 Open-ended generation task

As shown in Table 3, when applied alone, RePAIR outperforms most baselines across multiple metrics. For instance, it achieves the highest MAUVE score of 0.68 and the lowest CREP score of 1.83, indicating that it generates text more closely resembling real data with minimal degeneration. Moreover, it exhibits a distribution of unique $n$-gram scores comparable to that of the original WikiText-103 dataset. When combined with FOCUS or other baselines, RePAIR consistently improves all metrics while incurring only a slight increase in perplexity. On average, the MAUVE score improves by about 0.06 across methods, accompanied by a marked decrease in the CREP metric, indicating a significant reduction in repetitive generations. Notably, our approaches achieve both the highest MAUVE score and unique $n$-gram distributions that closely match real WikiText-103 text. Overall, these results demonstrate that our proposed methods substantially mitigate text

Table 4: Results of instruction-based generation on WikiText-103 dataset. Best per block in **bold**, second best underlined. CREP is reported as percentage values.

| Method | PPL ($\downarrow$) | 0-shot ($\uparrow$) | CREP ($\downarrow$) | EAD$_1$ ($\uparrow$) | BERTScore ($\uparrow$) |
|---|---|---|---|---|---|
| **LLaMA-3.1-8B-Instruct** | | | | | |
| KD | 25.65 | 62.32 | 1.97 | 0.28 | **0.49** |
| + UL | **25.52** | 62.01 | 3.10 | 0.28 | 0.48 |
| + SG | 25.53 | 62.30 | 2.23 | 0.28 | **0.49** |
| + DITTO | 25.59 | 61.97 | 1.30 | 0.28 | **0.49** |
| + RePAIR | 25.86 | **62.61** | **0.63** | **0.32** | **0.49** |
| FOCUS | 26.54 | **62.85** | 0.73 | 0.30 | **0.50** |
| + UL | 26.89 | 62.07 | 0.73 | 0.29 | 0.49 |
| + SG | **26.27** | **62.85** | 0.53 | 0.30 | **0.50** |
| + DITTO | 26.36 | 62.12 | 0.30 | 0.29 | **0.50** |
| + RePAIR | 26.20 | 62.61 | **0.23** | **0.31** | **0.50** |

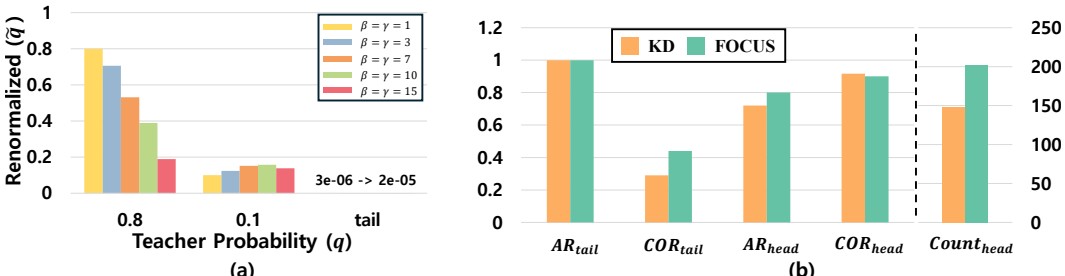

Figure 2: FOCUS analysis. (a) Renormalized distribution $\tilde{q}$ derived from the teacher probability $q$. (b) Comparison of token distributions between naive KD and FOCUS.

degeneration with only minimal trade-offs in perplexity, while generalizing effectively across both stand-alone and combined training strategies.

## 5.3 INSTRUCTION-BASED GENERATION TASK

For instruction-based generation, since the output length for each instruction may vary across methods, we use EAD$_1$ as the primary diversity metric instead of unique $n$-gram diversity, as the latter does not account for differences in generation length. As illustrated in Table 4, without FOCUS, the model achieves the lowest degeneration performance despite recording the best perplexity among baselines. In contrast, RePAIR attains a lower CREP score of 0.63 and the highest EAD$_1$ score of 0.32, while maintaining a comparable perplexity to UL. This indicates that RePAIR produces the most diverse generations while preserving semantic fidelity, as reflected in BERTScore. When applied with FOCUS, all of the methods show consistent improvements across metrics with only a slight increase in perplexity. For example, it achieves lower CREP, higher EAD$_1$, and improved BERTScore, while maintaining stable zero-shot accuracy. This indicates that the method mitigates degeneration and enhances generation quality without incurring knowledge loss. Notably, when FOCUS is combined with RePAIR, most metrics achieve the best results among all methods. Overall, these findings demonstrate that our proposed approaches not only mitigate degeneration more effectively than existing baselines but also preserve general knowledge and semantic quality in instruction-following tasks.

## 6 ANALYSIS

**FOCUS gradient study**  To demonstrate the effects of FOCUS, we consider a toy distribution where the top two tokens have probabilities of 0.8 and 0.1, with the remaining mass assigned to the tail. This setup reflects the typical Zipfian nature of token distributions, where a few high-ranked tokens dominate the probability mass while the rest form a long tail. We then plot both the original

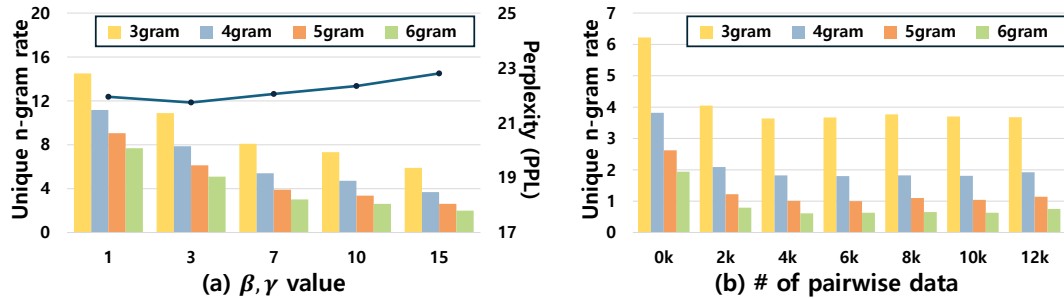

Figure 3: Ablation study of our methods on open-ended generation. (a) Unique $n$-gram with varying $\beta, \gamma$ values. (b) Unique $n$-gram with different amounts of pairwise data.

distribution $q$ and the reweighted distribution $\tilde{q}$ for different values of $\beta$ and $\gamma$. As shown on the left side of Figure 2-(a), $\tilde{q}$ provides the student with a smoother distribution in regions where the model is confident while preserving the relative relationships among token probabilities, and assigns near-zero probability to regions it intends to exclude. This prevents excessive concentration on specific tokens by scaling down their probabilities, thereby encouraging lexical diversity and reducing the positive feedback from prior context. Moreover, by increasing the probability mass in the tail, the model retains the possibility of selecting rare tokens, which further enhances diversity.

**FOCUS distribution study** As discussed in Section 3, our method encourages the student to follow both the tail and head regions of the teacher distribution. To compare these distributions, we compute the agreement rate and correlation. The agreement rate(AR) is defined as $\text{AR}_{\text{head}} = \frac{|H_q \cap H_p|}{|H_q|}$, $\text{AR}_{\text{tail}} = \frac{|T_q \cap T_p|}{|T_q|}$, while correlation(COR) is measured using Pearson's coefficient. In other words, we examine both whether the student follows the teacher's choices and the relative pattern of teacher distribution distributions. For this analysis, we first generate 200 samples from the teacher model and then feed the same set of samples into each method-applied student model for evaluation. The head region is determined using top-$p$ region with $p = 0.9$.

As shown in Figure 2-(b), the agreement in the tail region remains comparable between methods, whereas a clear difference appears in the correlation. This indicates that weighted training with the focus objective enables the student to better capture the teacher's tail patterns. In the head region, correlation slightly decreases, but agreement remains high. This suggests that the focus objective does not simply flatten the head distribution, but rather preserves the teacher's structure while smoothing it, thereby offering more plausible candidates. The effect is reflected in the increased number of average candidates in the head region, as indicated by $\text{count}_{head}$ $(148.6 \rightarrow 202.1)$.

**Hallucination Analysis** Following the results in Tab. 3 and Tab. 4, we observe a consistent upward trend in PPL when the proposed methods are applied. This suggests that the model's likelihood landscape becomes comparatively less stable, potentially weakening its factual grounding and increasing the risk of hallucination. To assess this aspect, we evaluate the Llama-3.1-8B models trained under each method on the TruthfulQA benchmark. Interestingly, as shown in Fig. 4, while FOCUS yields a consistent increase in PPL on the Wiki domain, it simultaneously improves performance on TruthfulQA. Among all methods, RePAIR yields the largest improvement in overall scores. We attribute this to its design: unlike the other approaches, RePAIR directly guides the student toward the spe-

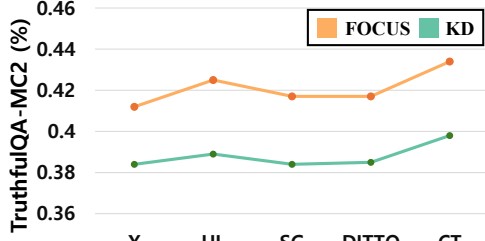

Figure 4: TruthfulQA-MC2 score. X indicates that no repetition-suppression technique was applied.

cific tokens preferred by the teacher during generation, providing a more explicit supervisory signal. Also, FOCUS achieves higher overall TruthfulQA scores than vanilla KD. This observation is consistent with prior findings that distillation is more effective when the transfer process accounts for

the reduced capacity of student or pruned models, directly following the teacher's target distribution (Wu et al., 2024; Yao et al., 2025a; Ko et al., 2024).

**Token-Level vs. Sequence-Level Signal**
We first note that RePAIR shares the general pairwise training structure commonly used in preference-based learning. The model is trained with positive and negative examples, and the objective encourages the preferred sample to be scored higher. A representative method following this structure is Direct Preference Optimization (DPO) (Rafailov et al., 2024). Although they share a similar high-level structure, their supervision signals differ in meaningful ways. RePAIR provides token-level corrective alternatives, offering explicit substitute tokens at the point where repetition begins. In contrast, DPO supplies only a sequence-level preference signal, encouraging the model to favor positive sentences overall without indicating where or how repetition should be corrected. To assess whether sequence-level preference learning can also mitigate repetition, we train DPO on the LLaMA-3.1-8B model using the same positive (non-repetitive) and negative (repetitive) samples used for RePAIR. Following the formulation in the original implementation (Rafailov et al., 2024), the DPO objective is defined as $\mathcal{L}_{\text{DPO}} = -\log \sigma \left( \beta_{dpo} \left( \Delta_{\text{student}} - \Delta_{\text{teacher}} \right) \right)$, where $\Delta_{\text{student}} = \log \pi_{\text{student}}(y^+ \mid x) - \log \pi_{\text{student}}(y^- \mid x)$ and $\Delta_{\text{teacher}} = \log \pi_{\text{teacher}}(y^+ \mid x) - \log \pi_{\text{teacher}}(y^- \mid x)$ with $\beta_{dpo}$ fixed to 0.1. We evaluate both RePAIR and DPO under the same WikiText-103 continuation setup used in Table 3. As illustrated in Table 5, the results show that DPO does reduce repetition, likely because the model learns to favor shorter continuations over longer repetitive ones. However RePAIR achieves a comparatively larger reduction in repetition and produces higher-quality generations, as reflected in the MAUVE scores (0.71 vs. 0.78). These findings support the interpretation that token-level corrective signals provide more effective guidance for mitigating degeneration than sequence-level preference optimization.

| Method | PPL ($\downarrow$) | Unique $n$-gram | | | |
|--------|-----|------|------|------|------|
| | | $n=3$ | $n=4$ | $n=5$ | $n=6$ |
| CE | **23.05** | 12.88 | 9.68 | 7.77 | 6.56 |
| DPO | 23.29 | 9.76 | 6.53 | 4.64 | 3.45 |
| RePAIR | 23.72 | **8.28** | **5.53** | **3.97** | **3.01** |

Table 5: Repetition comparison of DPO and RePAIR under the WikiText-103 continuation setup. For a clear comparison, KD and FOCUS are not applied here, and CE is used as the baseline.

**FOCUS trade-off** We conduct an ablation study on the hyperparameter $\beta, \gamma$ of FOCUS. We generate outputs on the open-ended WikiText-103 generation task described above and evaluate them using the unique $n$-gram metric. As shown in Figure 3-(a), increasing parameters consistently improves the overall unique $n$-gram, indicating that the model produces more diverse sentences. However, larger values of them slightly increase perplexity, since they encourage the model to rely more on high-confidence tokens during the distillation process. Nevertheless, as noted in Sections 5.2, higher values lead to improvements in generation-related metrics, suggesting that perplexity alone should not be regarded as the sole criterion for evaluating model quality.

**The number of pairwise data** In our experiments, we collected 12k pairwise samples for training and analyzed how many pairwise samples are required to reduce text degeneration. So, we trained As shown in Figure 3-(b), approximately 4k pairwise samples are sufficient to achieve repetition rate comparable to that obtained with the full 12k samples. This demonstrates that RePAIR is more data-efficient than DITTO, which requires half of the training set for its training loss function. Consequently, our approach leaves more data available for standard training, allowing the model to acquire broader knowledge from various data.

## 7 CONCLUSION

In this paper, we demonstrated that pruned models exhibit higher repetition rates compared to their unpruned counterparts, and that this phenomenon can be alleviated through token-level guidance strategies. To address this, we proposed FOCUS, a token probability weighted distillation approach that focuses on the high-confident token and RePAIR, a pairwise margin-based training objective that contrasts positive and negative samples guiding model to generate probable alternative tokens. Our experimental results show that across multiple evaluation metrics, the proposed methods effectively reduce repetition rates and enable the model to generate more accurate and coherent sentences in both open-ended and directed generation settings.

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

# A    GRADIENT ANALYSIS OF FOCUS

In this section, we analyze why FOCUS has effects on the suppressing repetition loop.

**Gradient Analysis of Knowledge distillation.**    The Knowledge Distillation (KD) loss is defined as the Kullback–Leibler (KL) divergence between the teacher distribution $q$ and the student distribution $p$:

$$L_{\text{KD}} = \sum_i q_i \log \frac{q_i}{p_i}. \tag{2}$$

Differentiating with respect to the softmax input $a$ (logit), we compute:

$$\frac{\partial L_{\text{KD}}}{\partial a_k} = -\sum_i q_i \frac{\partial \log p_i}{\partial a_k}.$$

The derivative of the log-softmax is:

$$\frac{\partial \log p_i}{\partial a_k} = \delta_{ik} - p_k,$$

where $\delta_{ik}$ is the Kronecker delta. Substituting back, we get:

$$\frac{\partial L_{\text{KD}}}{\partial a_k} = -q_k + \sum_i q_i p_k.$$

Since $\sum_i q_i = 1$, this simplifies to:

$$\frac{\partial L_{\text{KD}}}{\partial a_k} = p_k - q_k. \tag{3}$$

We can express it to form:

$$\nabla_a L_{\text{KD}} = p - q. \tag{4}$$

**Gradient Analysis of FOCUS**    Modifying the KL, FOCUS defines loss function as:

$$L_{\text{FOCUS}} = \sum_i w(q_i)\, q_i \log \frac{q_i}{p_i} \tag{5}$$

$$= \underbrace{\sum_i w(q_i)\, q_i \log q_i}_{\text{constant (independent of } a)} - \sum_i w(q_i)\, q_i \log p_i. \tag{6}$$

Differentiating $L_{\text{FOCUS}}$ with respect to $p_i$, we obtain:

$$\frac{\partial L}{\partial p_i} = -\frac{w_i q_i}{p_i}. \tag{7}$$

To compute the derivative with respect to the logits $a$, we need the softmax Jacobian:

$$\frac{\partial p_i}{\partial a_k} = p_i\,(\delta_{ik} - p_k), \tag{8}$$

where $\delta_{ik}$ is the Kronecker delta. By the chain rule, for each coordinate $k$:

$$\frac{\partial L}{\partial a_k} = \sum_i \frac{\partial L}{\partial p_i} \frac{\partial p_i}{\partial a_k} \tag{9}$$

$$= \sum_i \left( -\frac{w_i q_i}{p_i} \right) p_i \left( \delta_{ik} - p_k \right). \tag{10}$$

$$= \sum_i \left( -w_i q_i \right) \left( \delta_{ik} - p_k \right). \tag{11}$$

Next, we separate the summation into the cases $i = k$ and $i \neq k$:

$$\frac{\partial L}{\partial a_k} = \left( -w_k q_k \right) \left( 1 - p_k \right) \ + \ \sum_{i \neq k} \left( -w_i q_i \right) \left( -p_k \right). \tag{12}$$

Finally, we obtain:

$$\frac{\partial L}{\partial a_k} = -w_k q_k \ + \ p_k \sum_i w_i q_i. \tag{13}$$

Let

$$Z := \sum_j w_j q_j > 0 \tag{14}$$

Then the gradient can be written as

$$\frac{\partial L}{\partial a_k} = Z p_k - w_k q_k. \tag{15}$$

Now, define a reweighted teacher distribution $\tilde{q}$ as

$$\tilde{q}_k \quad \frac{w_k q_k}{Z}. \tag{16}$$

Substituting, we obtain

$$\frac{\partial L}{\partial a_k} = Z \left( p_k - \tilde{q}_k \right). \tag{17}$$

In vector form, the gradient of FOCUS loss is:

$$\nabla_a L_{\text{FOCUS}} \ = \ Z \left( p - \tilde{q} \right), \quad \text{where } \tilde{q} = \frac{w(q) \odot q}{\sum_j w(q_j) q_j}, \tag{18}$$

To this end, FOCUS can be viewed as optimizing the student distribution with respect to a reweighted teacher distribution.

## B  AN EXAMPLE OF PAIRWISE DATA

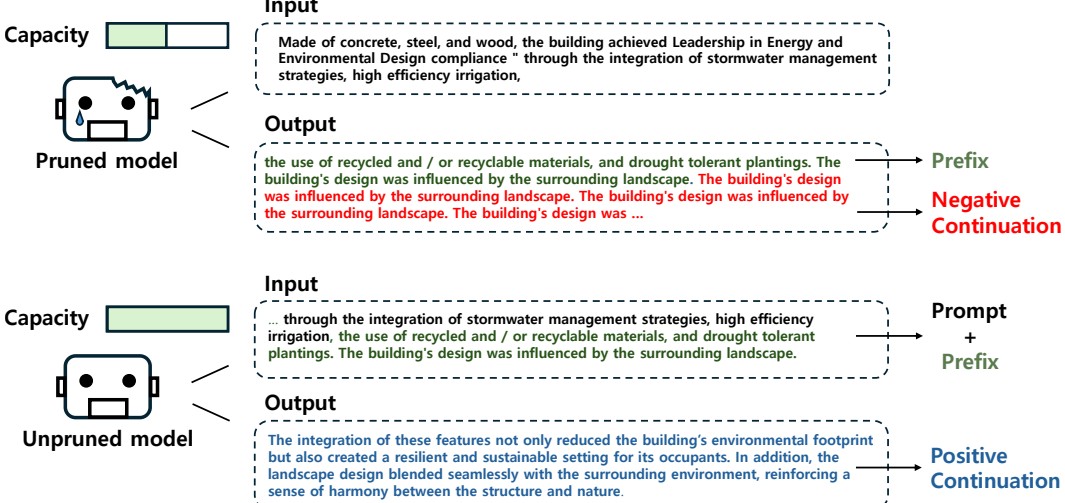

Figure 5: Example of pairwise data construction.

As illustrated in Figure 5, we first provide an input prompt to the pruned model and identify the onset of degeneration using the coverage metric. The sequence up to this point is designated as the prefix. We then feed the same input along with the prefix to the unpruned model to obtain a positive continuation that remains free of degeneration.

Finally, we construct training pairs of the form (prompt + prefix, negative continuation) and (prompt + prefix, positive continuation) to optimize the margin loss. In total, we collect 12k pairwise data. But, as shown in Section 3, even 4k pairs are sufficient, highlighting the cost-effectiveness of the approach.

# C ZEROSHOT ACCURACY

Table 6: Zero-shot evaluation results on the LLaMA-3.1-8B model across six benchmarks. The best result within each block is highlighted in **bold**, while the second best is underlined.

| Model | Arc_c | PIQA | BoolQ | OpenQA | Hellaswag | Winogrande | Average |
|---|---|---|---|---|---|---|---|
| KD | 43.26 | 77.58 | 67.52 | 40.00 | 71.05 | 62.90 | 60.39 |
| + UL | **44.37** | 77.37 | 69.60 | **41.20** | 71.08 | 63.30 | 61.15 |
| + SG | 43.52 | 77.58 | 69.69 | 40.80 | 71.11 | 62.59 | 60.88 |
| + DITTO | 43.43 | 77.42 | 69.27 | 40.40 | 70.85 | 62.35 | 60.62 |
| + RePAIR | 43.17 | **77.86** | **70.55** | **41.20** | **71.86** | **63.54** | **61.36** |
| FOCUS | 42.92 | 77.26 | 65.87 | 40.20 | 70.70 | 63.22 | 60.03 |
| + UL | 43.34 | 77.15 | 67.55 | 40.80 | 70.42 | **63.46** | 60.45 |
| + SG | 42.92 | 77.20 | 68.17 | **41.00** | 70.69 | 62.83 | 60.47 |
| + DITTO | 42.66 | 77.15 | 69.08 | 40.80 | 70.46 | 63.06 | 60.54 |
| + RePAIR | **43.52** | **77.42** | **69.79** | 40.00 | **71.60** | 62.83 | **60.86** |

Table 7: Zero-shot evaluation results on the LLaMA23-13B model across six benchmarks. The best result within each block is highlighted in **bold**, while the second best is underlined.

| **Model** | **Arc_c** | **PIQA** | **BoolQ** | **OpenQA** | **Hellaswag** | **Winogrande** | **Average** |
|---|---|---|---|---|---|---|---|
| KD | 45.65 | 77.64 | 73.12 | **44.20** | 75.98 | 68.03 | 64.10 |
| + UL | **46.59** | 77.75 | 73.15 | 43.80 | 76.05 | **68.35** | 64.28 |
| + SG | 45.90 | 77.69 | 73.15 | 44.00 | 76.07 | 68.11 | 64.15 |
| + DITTO | 45.48 | 77.64 | 72.91 | 43.20 | 75.74 | 68.11 | 63.85 |
| + RePAIR | **46.59** | **77.75** | 72.78 | **44.20** | **76.80** | 68.03 | **64.36** |
| FOCUS | 45.99 | 77.26 | 73.15 | 43.20 | 74.98 | 67.56 | 63.69 |
| + UL | 45.90 | 77.31 | 72.97 | 43.00 | 74.68 | 67.56 | 63.57 |
| + SG | 45.82 | **77.48** | **73.33** | 43.20 | 75.00 | **67.80** | 63.77 |
| + DITTO | 45.65 | 77.26 | 72.72 | 43.20 | 74.66 | 67.64 | 63.52 |
| + RePAIR | **46.16** | 77.26 | 72.42 | **43.80** | **76.27** | 67.56 | **63.91** |

Table 8: Zero-shot evaluation results on the LLaMA-3.1-8B-Instruct model across six benchmarks. The best result within each block is highlighted in **bold**, while the second best is underlined.

| Model | Arc_c | PIQA | BoolQ | OpenQA | Hellaswag | Winogrande | Average |
|---|---|---|---|---|---|---|---|
| KD | 45.39 | 78.02 | 73.82 | 39.60 | 70.87 | 66.22 | 62.32 |
| + UL | 45.39 | 77.75 | 73.18 | 39.00 | 70.84 | 65.90 | 62.01 |
| + SG | **45.65** | 78.07 | 73.21 | 39.40 | **70.95** | 66.54 | 62.30 |
| + DITTO | 44.37 | 77.97 | 72.14 | **40.00** | 70.78 | 66.30 | 61.93 |
| + FOCUS | 45.48 | **78.13** | **73.67** | 39.80 | 71.71 | **66.85** | **62.61** |
| FOCUS | 45.39 | **78.62** | 73.85 | **42.20** | 71.00 | **66.06** | **62.85** |
| + UL | **46.08** | 77.58 | 72.05 | 40.80 | 70.95 | 64.96 | 62.07 |
| + SG | 45.82 | 77.70 | **74.04** | **42.20** | 71.05 | 66.30 | **62.85** |
| + DITTO | 44.76 | 77.86 | 71.35 | 41.60 | 70.98 | 66.14 | 62.12 |
| + RePAIR | 45.56 | 77.74 | 73.52 | 41.80 | **71.75** | 65.27 | 62.61 |

**Evaluation Tasks**    We conduct zero-shot evaluations on the following benchmark tasks:

- **ARC_c**   (AI2 Reasoning Challenge): A multiple-choice science exam dataset that evaluates complex reasoning ability beyond simple fact recall.

- **PIQA**   (Physical Interaction Question Answering): A benchmark for testing physical commonsense reasoning, where models must choose the more plausible solution to everyday tasks.

- **BoolQ**   (Boolean Questions): A reading comprehension dataset consisting of yes/no questions with corresponding passages from Wikipedia.

- **OpenQA** (Open-domain Question Answering): A task that requires answering factoid questions based on open-domain knowledge, without access to a fixed context passage.

- **Hellaswag**: A commonsense reasoning benchmark where the model must select the most plausible continuation of a given context.

- **Winogrande**: A large-scale pronoun resolution dataset designed to test commonsense reasoning through fill-in-the-blank style questions.

# D    IMPLEMENTATION DETAILS

In this section, we leave the implementation detail of baseline methods. All of the methods are implemented using huggingface framework and based on the official implementation.

**Knowledge Distillation**    Knowledge distillation (KD) is a generally usde technique to transfer knowledge from a large teacher model to a smaller student model.

$$L_{\text{KD}} = T^2 \cdot \text{KL}\left( \text{softmax}\left(\tfrac{z_t}{T}\right) \,\Big\|\, \text{softmax}\left(\tfrac{z_s}{T}\right) \right), \tag{19}$$

where $z_t$ and $z_s$ denote the logits of the teacher and student, respectively, and $T$ is the temperature parameter that controls the smoothness of the distributions. In our experiments, we adopt a temperature of $T = 2$, which is generally used and provides a good balance between stable training and effective knowledge transfer.

**Unlikelihood Training**    Unlikelihood training aims to reduce the probability of generating repeated tokens by penalizing candidates that already appear in the previous context. Although sentence-level variants have been explored in prior work with smaller models, applying them to large-scale models such as LLaMA is challenging due to memory constraints. Therefore, we adopt a token-level variant.

Specifically, at step $t$, we define the set of negative candidates as

$$C^t_{\text{prev-context}} = \{x_1, \ldots, x_{t-1}\} \setminus \{x_t\}. \tag{20}$$

To combine UL loss with maximum likelihood training, we adopt a token-level objective:

$$\mathcal{L}^t_{\text{UL-token}}(p_\theta(\cdot \mid x_{<t}), C^t) = -\alpha \cdot \sum_{c \in C^t} \log\left(1 - p_\theta(c \mid x_{<t})\right) \ - \ \log p_\theta(x_t \mid x_{<t}), \tag{21}$$

In our experiments, we set $\alpha = 0.5$, which provides a balanced trade-off between aggressively suppressing repetitions and preserving overall fluency and perplexity.

**ScaleGrad**    ScaleGrad is a repetition-penalization method that rescales the gradient on specific tokens which appeared previously in the context.

$$\tilde{p}_i = \begin{cases} \dfrac{\gamma \cdot p_i}{\sum_{j=1}^{|\mathcal{S}_{\text{novel}}|} \gamma \cdot p_j + \sum_{j=1}^{|\mathcal{V}'|} p_j}, & \text{if } i \in \mathcal{S}_{\text{novel}}, \\[4mm] \dfrac{p_i}{\sum_{j=1}^{|\mathcal{S}_{\text{novel}}|} \gamma \cdot p_j + \sum_{j=1}^{|\mathcal{V}'|} p_j}, & \text{otherwise.} \end{cases}$$

A smaller value of $\gamma$ more aggressively suppresses repetition, but at the cost of deteriorating perplexity. In the original paper, the authors experimented with $\gamma \in \{0.2, 0.5, 0.8\}$, and selected the value for each task accordingly. In our experiments, we set $\gamma = 0.5$, as it achieves a good balance between mitigating repetition and preserving perplexity.

**DITTO**    The authors focus on the self-reinforcement effect at the sentence level. To penalize repetition, they construct pseudo-repetition sentences and define the loss function as follows:

$$\mathcal{L}^{n,l}_{\text{DITTO}}\big(\mathcal{P}_\theta(x_{n,l} \mid \mathbf{x}_{<n,l})\big) = -\log\Big(1 - \big|\mathcal{P}_\theta(x_{n,l} \mid \mathbf{x}_{<n,l}) - \lambda \cdot \mathcal{P}^*_\theta(x_{n-1,l} \mid \mathbf{x}_{<n-1,l})\big|\Big).$$

Following the baseline setting, we use half of the training data for pseudo-repetition penalization. To construct pseudo-repetition data, the instruction and input are taken as a prefix, and the output is repeatedly appended until the maximum sequence length is reached. In accordance with the original implementation, we employ an MSE loss with $\lambda = 0.5$.

# E PARAMETER SEARCH

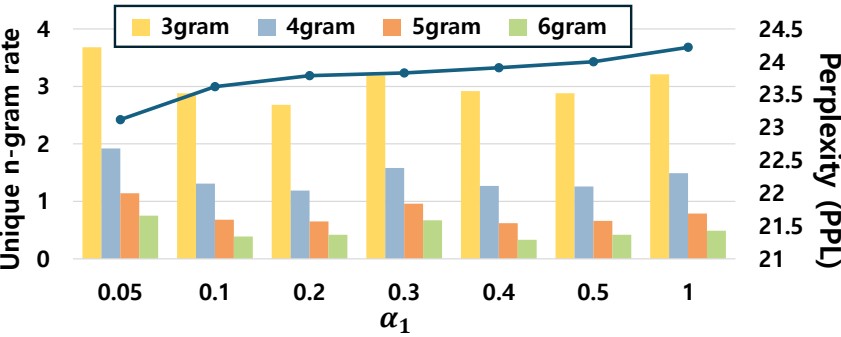

Figure 6: Hyperparameter search for $\alpha_1$

As shown in Figure 3, FOCUS influences perplexity (PPL) as $\beta$ increases, making the choice of $\gamma_1$ crucial for controlling model perplexity. Following the main experiments, we fix $\beta$ at 15. To tune $\alpha_1$, we compute PPL and track the number of unique n-grams as its value increases. As shown in Figure 6, PPL deteriorates sharply when $\alpha_1$ is increased from 0.05 to 0.1, while the n-gram rate saturates from 0.1 onward. Therefore, we set $\alpha_1 = 0.05$ to balance PPL and $n$-gram diversity.

# F CREP: COVERAGE-BASED REPETITION METRIC

To more rigorously detect text degeneration, we evaluate repetition across a broad range of $n$-gram lengths. For each generated sentence, we sweep $r$ from $4$ to $16$ and identify the most frequently recurring $r$-gram. We then reconstruct the full repeated segment and measure how much of the remaining output it covers globally. A sentence is marked as degenerate if its maximum coverage across all $r$ exceeds a predefined threshold. The full procedure is summarized in Algorithm 1.

---

**Algorithm 1:** CREP: Coverage-based Repetition Detection

---

**Input:** Dataset of generated texts $\mathcal{D}$, $n$-gram range $[r_{\min}, r_{\max}]$, global coverage threshold $\theta$
**Output:** CREP score in $[0, 1]$

---

1   $deg\_count \leftarrow 0$
2   **foreach** $y \in \mathcal{D}$ **do**
3     $t \leftarrow \text{TOKENIZE}(y)$
4     $best\_cov \leftarrow 0$
5     **for** $r = r_{\min}$ **to** $r_{\max}$ **do**
6       **if** $|t| < r + 1$ **then**
7         **continue**
8       $(g^{\star}, pos) \leftarrow \text{NGRAM\_WITH\_POSITIONS}(t, r)$
9       **if** $|pos| < 2$ **then**
10        **continue**
11       $\Delta \leftarrow \{pos_{i+1} - pos_i \mid i = 1, \ldots, |pos| - 1\}$
12       $d^{\star} \leftarrow \text{MODE}(\Delta)$
13       Find smallest $i$ such that $pos_{i+1} - pos_i = d^{\star}$
14       $s_1 \leftarrow pos_i$
15       $s_2 \leftarrow s_1 + d^{\star}$
16       $p \leftarrow t[s_1 : s_2]$            ▷ candidate repeated segment
17       $u \leftarrow t[s_2 : |t|]$          ▷ tail after the first repetition
18       $cov\_full \leftarrow \text{GLOBALCOVERAGE}(p, u, t)$
19       $best\_cov \leftarrow \max(best\_cov, cov\_full)$
20     **if** $best\_cov \geq \theta$ **then**
21       $deg\_count \leftarrow deg\_count + 1$
22   **return** $CREP = deg\_count / |\mathcal{D}|$

---

# G    AGGRESSIVE PRUNING SETTINGS

Table 9: Results at 35% and 45% pruning using LLMPruner. Lower is better for PPL and $n$-gram repetition; higher is better for MAUVE.

| Method | PPL ($\downarrow$) | 3-gram ($\downarrow$) | 4-gram ($\downarrow$) | 5-gram ($\downarrow$) | 6-gram ($\downarrow$) | MAUVE ($\uparrow$) |
|--------|------|--------|--------|--------|--------|-------|
| | | | **LLaMA-3.1-8B (35%)** | | | |
| FOCUS | **26.26** | 5.76 | 3.45 | 2.33 | 1.73 | 0.72 |
| + UL | 26.80 | 5.06 | 2.97 | 2.00 | 1.48 | 0.71 |
| + SG | 26.39 | 5.21 | 3.10 | 2.13 | 1.59 | **0.73** |
| + DITTO | 26.72 | 4.50 | 2.36 | 1.40 | 0.89 | 0.70 |
| + RePAIR | 27.12 | **3.22** | **1.46** | **0.75** | **0.43** | 0.70 |
| | | | **LLaMA-3.1-8B (45%)** | | | |
| FOCUS | **33.27** | 6.87 | 4.46 | 3.26 | 2.55 | 0.44 |
| + UL | 33.94 | 5.55 | 3.42 | 2.37 | 1.79 | 0.34 |
| + SG | 33.36 | 6.18 | 4.05 | 2.97 | 2.36 | 0.44 |
| + DITTO | 34.12 | 4.46 | 2.52 | 1.60 | 1.09 | 0.35 |
| + RePAIR | 34.12 | **3.40** | **1.62** | **0.89** | **0.55** | **0.46** |

To demonstrate whether the proposed methods remain effective under more aggressive sparsity, we additionally evaluate models pruned at 35% and 45% using LLMPruner. Prior work has shown that pruning ratios above 30% typically induce noticeable degeneration (Jaiswal et al., 2024). Therefore, these settings allow us to assess the robustness of our approach in regimes where degradation is more prominent. As illustrated in Table 9, higher pruning ratios lead to increased perplexity and reduced MAUVE, reflecting the reduced capacity of the compressed models. Nevertheless, both FOCUS and RePAIR consistently mitigate excessive repetition across all sparsity levels.

