# OpenReview forum: "Mitigating Text Degeneration via Token-Level Guidance For pruned Large Language Models"
_ICLR.cc/2026/Conference — Submitted to ICLR 2026_

### Official Review · Reviewer_7Xnn · 2025-10-29

**Soundness:** 2
**Presentation:** 3
**Contribution:** 2
**Rating:** 4
**Confidence:** 3

**Summary:**

The paper proposes two strategies to mitigate repetitive degeneration in pruned model outputs.
The first (FOCUS) redistributes probability mass over the top-probability tokens during the model distillation phase, moving more probability mass onto less probable tokens.
The second method, PT, uses a contrastive objective to get the model to generate text more like the teacher model, and less like degenerate outputs from the pruned model.
Along the way, the authors introduce a new metric, CREP, to measure repetitive degeneration in model outputs.

**Strengths:**

The paper focuses on an important problem. The method appears to be effective in reducing repetitiveness. The writing is clear, with the exception of the explanation of the FOCUS objective, which lack motivation in my opinion.

**Weaknesses:**

1. I am somewhat unsatisfied with the FOCUS objective's motivation, explanation, and intuition. Looking at the weighting function, it seems to give full weight (1) to tokens close to 0 and 1 probability, and has a u-shape, giving the least probability to tokens near 0.5 probability. It is not clear why this "gives more weight to high confidence tokens" as you say, and if it does, why use this particular formulation? Figure 2a seems to demonstrate that the particular distribution becomes more high-entropy. If that is the goal, why not adjust the teacher distribution with something more standard, like temperature? If temperature does not work as well, why? It would be good to see some ablations.

2. The paper you cite (Jaiswal 2024) showing degeneration in modern LLMs only shows significant degeneration _after_ 30% pruning and in your experiments, you use only 25% pruning. Furthermore, language is naturally repetitive to some degree. The lack of baselines for repetition metrics for unpruned models makes it unclear whether the method is needlessly removing the natural level of repetition in the generations.

3. Overall, my excitement for the paper is middling. It does not seem to offer significant insights that broaden our understanding of language models, degeneration, or distillation. Rather, it offers two engineering ideas that they argue works well in practice. PT seems like the better motivated of the two ideas, but it does not in my mind break new ground. Contrastive objectives to make the student model more like the teacher seems like an obvious idea and I would be surprised if it is not already used in practice.

**Questions:**

1. Why did you think it necessary to define a new repetition metric? Why are existing metrics insufficient? For instance the one used in the nucleus sampling paper (Holtzman 2019).

2. Your contrastive learning objective has some similarities in terms of inputs (favored/disfavored example pairs) to DPO. Did you experiment with DPO? How would you characterize the difference/connection?

Typos
- 465: thm

---

> ### Author Response · Authors · 2025-11-20
> **Response to Reviewer 7Xnn (part 1)**
>
> Thank you for your detailed comments. We address the reviewer’s questions here:
>
> ---
> > Why did you think it necessary to define a new repetition metric? Why are existing metrics insufficient? For instance the one used in the nucleus sampling paper (Holtzman 2019).
>
> Degeneration caused by repetition typically occurs in the latter part of a generated sentence. Because of this, a metric such as unique n-gram can be diluted if the model produces sufficiently varied tokens at the beginning of the sequence, making it difficult for the metric to fully capture the presence of degeneration. In instruction-based generation, the output lengths vary considerably, which further complicates the use of unique n-gram as a sole indicator. Although EDA provides a numerical measure of token-level diversity, it does not intuitively reflect how severe the degeneration is.
>
> For these reasons, we view these metrics as complementary rather than sufficient on their own. In our paper, we adopt CREP, which evaluates repetition more rigorously by checking, for each sentence, whether n-grams dominate more than 30% of the sentence across a range of n from 4 to 16. This allows for a stricter and more intuitive assessment of degeneration. We apologize for not describing the CREP computation in sufficient detail in the main text.
>
> ---
> > Your contrastive learning objective has some similarities in terms of inputs (favored/disfavored example pairs) to DPO. Did you experiment with DPO? How would you characterize the difference/connection?
>
> PT and DPO share a similar structure in that both learn from positive and negative pairs. PT identifies the points at which the model begins to repeat and directly provides alternative tokens that should be generated, offering explicit token-level guidance at given point. DPO, in contrast, encourages the model to prefer the overall style of positive sentences rather than prescribing specific token substitutions.
>
> As the reviewer suggested, encouraging preference toward non-repetitive sentences via DPO may also help reduce degeneration. To examine this possibility, we trained DPO on the Llama-3.1-8B model using positive samples without repetition and negative samples containing repetition. The results are reported in the table below. (Both PT and DPO were trained using the same amount of positive and negative samples.) The evaluation was conducted using the same Wiki continuation setup as in Table 1 of the paper.
>
> | Method | PPL   | 3-gram | 4-gram | 5-gram | 6-gram | MAUVE |
> |--------|-------|--------|--------|--------|--------|--------|
> | CE     | 23.05 | 12.88  | 9.68   | 7.77   | 6.56   | 0.74   |
> | DPO    | 23.29 | 9.76   | 6.53   | 4.64   | 3.45   | 0.71   |
> | PT     | 23.72 | 8.28   | 5.53   | 3.97   | 3.01   | 0.78   |
>
> According to the Table, DPO is also able to reduce repetition. This appears to stem from the model learning to prefer shorter, non-repetitive sentences over longer sentences that contain repetition. Nevertheless, PT, which provides direct token-level guidance at the points where repetition emerges, achieves a greater reduction in repetition and produces higher-quality outputs, as reflected in the MAUVE scores (0.71 vs. 0.78).
>
>
> ---
> >  Typos - 465: thm
>
> Thank you for notifying typos. I will revise the sentence in the text.

---

> ### Author Response · Authors · 2025-11-20
> **Response to Reviewer 7Xnn (part 2)**
>
> > The paper you cite (Jaiswal 2024) showing degeneration in modern LLMs only shows significant degeneration after 30% pruning and in your experiments, you use only 25% pruning. Furthermore, language is naturally repetitive to some degree. The lack of baselines for repetition metrics for unpruned models makes it unclear whether the method is needlessly removing the natural level of repetition in the generations.
>
> To demonstrate that our methods remain effective even under high pruning ratios, we conducted additional experiments by applying 35% and 45% pruning using LLMPruner. These experiments were performed on the Llama-3.1-8B model.
>
> ### Results at 35% Pruning
> | Method    | PPL   | 3-gram | 4-gram | 5-gram | 6-gram | MAUVE |
> |-----------|-------|--------|--------|--------|--------|--------|
> | FOCUS     | 26.26 | 5.76   | 3.45   | 2.33   | 1.73   | 0.72   |
> | +UL       | 26.80 | 5.06   | 2.97   | 2.00   | 1.48   | 0.71   |
> | +SG       | 26.39 | 5.21   | 3.10   | 2.13   | 1.59   | 0.73   |
> | +DITTO    | 26.72 | 4.50   | 2.36   | 1.40   | 0.89   | 0.70   |
> | +PT       | 27.12 | 3.22   | 1.46   | 0.75   | 0.43   | 0.70   |
>
> ### Results at 45% Pruning
> | Method    | PPL   | 3-gram | 4-gram | 5-gram | 6-gram | MAUVE |
> |-----------|-------|--------|--------|--------|--------|--------|
> | FOCUS     | 33.27 | 6.87   | 4.46   | 3.26   | 2.55   | 0.44   |
> | +UL       | 33.94 | 5.55   | 3.42   | 2.37   | 1.79   | 0.34   |
> | +SG       | 33.36 | 6.18   | 4.05   | 2.97   | 2.36   | 0.44   |
> | +DITTO    | 34.12 | 4.46   | 2.52   | 1.60   | 1.09   | 0.35   |
> | +PT       | 34.12 | 3.40   | 1.62   | 0.89   | 0.55   | 0.46   |
>
> As shown in both tables, FOCUS and PT continue to mitigate repetition even at higher pruning levels. At the same time, increasing the pruning ratio leads to higher perplexity and lower MAUVE scores, which is a natural consequence of reduced model capacity under more aggressive sparsity.
>
> Moreover, as the reviewer noted, not all repetition is undesirable, since natural repetition can occur in coherent text. For this reason, we also computed the actual Unique n-gram statistics from randomly sampled WikiText-103 references. These values were reported in Table 1 of the main paper, and we showed that applying FOCUS and PT together preserves a similar level of natural diversity.
>
>
>
> ---
> > I am somewhat unsatisfied with the FOCUS objective's motivation, explanation, and intuition. Looking at the weighting function, it seems to give full weight (1) to tokens close to 0 and 1 probability, and has a u-shape, giving the least probability to tokens near 0.5 probability. It is not clear why this "gives more weight to high confidence tokens" as you say, and if it does, why use this particular formulation? Figure 2a seems to demonstrate that the particular distribution becomes more high-entropy. If that is the goal, why not adjust the teacher distribution with something more standard, like temperature? If temperature does not work as well, why? It would be good to see some ablations.
>
>
> The reason we emphasize the high-confidence regions (i.e., probabilities close to 0 or 1) is to address a well-known limitation of forward KL in distillation.
> KL is defined as $ \sum_x p_t(x)\,\log\left(\frac{p_t(x)}{p_s(x)}\right)$. KL assigns gradients in proportion to the teacher probability; therefore, tokens to which the teacher assigns very low probability receive almost no penalty. As a result, the student does not reliably suppress tokens that the teacher intends to exclude. This issue has been widely noted in prior work, which is why several studies have explored alternatives or modifications to KL, such as reverse KL.
>
> Moreover, as discussed in Section 3.2, a pruned student model does not have enough capacity to learn every aspect of the teacher distribution. For this reason, we intentionally sacrifice the mid-probability region, even though this can slightly increase perplexity, our goal is to place greater emphasis on tokens that the teacher clearly intends to generate (probabilities close to 1) or clearly intends to suppress (probabilities close to 0). By doing so, the student is more likely to retain valid candidates in the top-p or top-k region during sampling, while avoiding tokens that should not appear. This selective emphasis helps reduce repetition loops and leads to higher-quality generation.
>
> We would also like to note that the temperature-based approach suggested by the reviewer operates differently from the mechanism used in FOCUS. FOCUS does not modify the teacher distribution itself. Instead, it adjusts the weights of loss while keeping the original teacher probabilities intact. This allows the student to reference the teacher distribution as is, while selectively emphasizing the parts of the teacher distribution that are intentionally targeted for learning.

---

### Official Review · Reviewer_kxzd · 2025-10-31

**Soundness:** 3
**Presentation:** 3
**Contribution:** 2
**Rating:** 6
**Confidence:** 4

**Summary:**

When pruning LLM's language model capabilities often degenerate, to recover some post-training is applied, however artifacts like repetition-loops may persist. The authors propose to augment standard CE-loss with FOCUS, a teacher loss - weighted knowledge destillation, and PT, pairwise margin-based training. Experiments show that this indeed fosters diversity in generations.

**Strengths:**

- clear motivation and methodolgy and experiment and written form - overall solid work

**Weaknesses:**

W1 weighted KD and a ranking loss are well established methods, not surprising that it will improve (as long as there are fitting teacher/ samples), so the major novelty is applying it to pruning/ finding that it solves this particular error case. this error case again is very typical and not surprising that 'better training signal fixes it faster'. i'm missing ablations on when pure CE also fixes the repetition pitfall. given that this is ICLR main track, i cannot give it better than weak accept here.

W2 the focus on this paper is on diversity generation only. ppl and standard evaluation benchmarks consistently get slightly worse for basemodel. in instruction model benchmarks slightly improve but PPL still gets worse? i guess a paragraph / discussion on how meaningless the deviations in perplexity/ zero shot are could be in accordance to [1]

W3 [1] also shows that 25% pruning in a non-uniform fashion is actually pretty well achievable with almost no degradation. imho it would be necessary to see if the benefits persist on larger pruning rates/ different pruning techniques.


[1] https://arxiv.org/abs/2311.01544

**Questions:**

adressing /commenting weaknesses

---

> ### Author Response · Authors · 2025-11-20
> **Response to Reviewer kxzd**
>
> Thank you for your detailed comments. We addressed questions here:
>
> ---
> > W1 weighted KD and a ranking loss are well established methods, not surprising that it will improve (as long as there are fitting teacher/ samples), so the major novelty is applying it to pruning/ finding that it solves this particular error case. this error case again is very typical and not surprising that 'better training signal fixes it faster'. i'm missing ablations on when pure CE also fixes the repetition pitfall. given that this is ICLR main track, i cannot give it better than weak accept here.
>
> As the reviewer pointed out, an ablation showing that pure CE does not effectively reduce degeneration in pruned models is indeed important. Therefore, we trained a pruned model using only the CE loss under the same experimental settings described in the Experiment section, and the results are presented below:
>
> | Method | PPL   | 3-gram | 4-gram | 5-gram | 6-gram | MAUVE |
> |--------|-------|--------|--------|--------|--------|--------|
> | CE     | 23.05 | 12.88  | 9.68   | 7.77   | 6.56   | 0.74   |
>
> These findings confirm that, consistent with prior work, degeneration in pruned models is difficult to mitigate using CE alone. We will include this as an additional ablation in the revised version.
>
>
> ---
> > the focus on this paper is on diversity generation only. ppl and standard evaluation benchmarks consistently get slightly worse for basemodel. in instruction model benchmarks slightly improve but PPL still gets worse? i guess a paragraph / discussion on how meaningless the deviations in perplexity/ zero shot are could be in accordance to  [1]
>
> It would be beneficial to show that the slight increase in PPL observed across our experiments has only a marginal impact on overall model quality. An increase in perplexity can indicate potential instability in the model’s distribution, which in turn may elevate the risk of hallucination. To assess this aspect, we evaluated the Llama-3.1-8B models used in our experiments on the TruthfulQA-MC2 benchmark.
>
> | Method | X     | UL    | SG    | DITTO | CT    |
> |--------|-------|-------|-------|--------|-------|
> | **KD**     | 0.384 | 0.389 | 0.384 | 0.385 | 0.398 |
> | **FOCUS**  | 0.412 | 0.425 | 0.417 | 0.417 | 0.434 |
> * X denotes that KD/FOCUS is only applied.
>
> In contrast to the increase in PPL, applying FOCUS and CT improves the model’s ability to distinguish factual information. This result suggests that the rise in PPL has only a marginal impact on downstream quality.
>
> In addition, as shown in Table 1, although PPL becomes worse in most cases, the generation quality measured by MAUVE improves. This supports the idea that a higher PPL does not necessarily indicate degraded generation performance. A detailed discussion of this observation will be added to the analysis section.
>
> ---
> > W3 [1] also shows that 25% pruning in a non-uniform fashion is actually pretty well achievable with almost no degradation. imho it would be necessary to see if the benefits persist on larger pruning rates/ different pruning techniques.
>
> As the reviewer noted, degeneration generally becomes more severe as the pruning ratio increases. To investigate whether our method can still reduce repetition beyond the 25% pruning setting, we conducted additional experiments using the Llama-3.1-8B model with LLMPruner at higher sparsity levels, specifically 35% and 45%.
> ### Results at 35% Pruning
> | Method    | PPL   | 3-gram | 4-gram | 5-gram | 6-gram | MAUVE |
> |-----------|-------|--------|--------|--------|--------|--------|
> | FOCUS     | 26.26 | 5.76   | 3.45   | 2.33   | 1.73   | 0.72   |
> | +UL       | 26.80 | 5.06   | 2.97   | 2.00   | 1.48   | 0.71   |
> | +SG       | 26.39 | 5.21   | 3.10   | 2.13   | 1.59   | 0.73   |
> | +DITTO    | 26.72 | 4.50   | 2.36   | 1.40   | 0.89   | 0.70   |
> | +PT       | 27.12 | 3.22   | 1.46   | 0.75   | 0.43   | 0.70   |
>
> ### Results at 45% Pruning
> | Method    | PPL   | 3-gram | 4-gram | 5-gram | 6-gram | MAUVE |
> |-----------|-------|--------|--------|--------|--------|--------|
> | FOCUS     | 33.27 | 6.87   | 4.46   | 3.26   | 2.55   | 0.44   |
> | +UL       | 33.94 | 5.55   | 3.42   | 2.37   | 1.79   | 0.34   |
> | +SG       | 33.36 | 6.18   | 4.05   | 2.97   | 2.36   | 0.44   |
> | +DITTO    | 34.12 | 4.46   | 2.52   | 1.60   | 1.09   | 0.35   |
> | +PT       | 34.12 | 3.40   | 1.62   | 0.89   | 0.55   | 0.46   |
>
> As illustrated in both Table, FOCUS and PT remain effective in reducing repetition rates even under high levels of pruning. However, as the pruning ratio increases, we also observe a degradation in perplexity and a drop in MAUVE scores, which is expected given the worsening model capacity at higher sparsity levels.

---

### Official Review · Reviewer_3UTo · 2025-10-31

**Soundness:** 2
**Presentation:** 2
**Contribution:** 2
**Rating:** 2
**Confidence:** 4

**Summary:**

This paper proposes two methods for mitigating token repetition. The first, FOCUS, takes KL-divergence minimization to a stronger teacher, and changes the expectation to instead be weighted by a transformed function of the teacher’s likelihood – focusing more on very high or very low probabilities. The second, PT, generates synthetic preference data by first detecting student-generated examples of repetition (including the prefix where repetition starts), and then rolling out better continuations for that prefix from a stronger teacher model. These preference pairs are then used to finetune the student model using a loss that incorporates a hinge-like threshold to a log-likelihood preference loss.

The paper evaluates these methods in a setting in which the student model is low-quality due to pruning, and the teacher is the unpruned model.

The paper evaluates on various GLUE-like benchmarks (e.g., PIQA, BoolQ) as well as some open-ended generation (MAUVE and PPL, I think on wikitext 103-like text.)

**Strengths:**

This paper explores some interesting ideas – it has a nice repetition measurement heuristic. The method for detecting repetition and rolling out non-repetitive texts for preference learning is nice. Ideas around sharpening the KL-divergence are also interesting to explore. Overall, this paper proposes a bunch of potentially interesting ideas for getting any model that has repetition issues to behave more like a teacher model that doesn’t.

I think the ideas here have value, and a clarified version of this paper in which the individual aspects of each method are tested more clearly and the experimental setup were stronger would lead to a higher score.

**Weaknesses:**

Unfortunately, I have a variety of concerns about this work. I’ll give examples and places for improvement, but overall I found myself struggling to take clear conclusions from the work.

For all experiments, it seems like MAUVE, n-gram uniqueness, and PPL metrics are computed over Wikitext text after finetuning on instruction-response formatted data from Alpaca. This train-test mismatch does not make sense to me – why should we expect the model to generate wikitext-like articles or have low wikitext PPL after finetuning on chat-formatted instruction-response data?

For FOCUS for example, the hypotheses seems to be that KL-divergence regularization to a teacher doesn’t focus enough on tokens that are low-probability under the teacher (because the importance weight, the teacher’s probability, is close to 0.) KL-divergence does penalize putting probability mass on tokens that are low-probability under the teacher through the partition function of the softmax – the probabilities must sum to 1, so putting too-high probability on a low-probability token is penalized. The hypothesis here is unclear to me because we don’t see evidence in this paper that the particular weighting of KL is bad in this setting (the pruning-recovery setting) beyond FOCUS leading to higher scores under the authors’ proposed repetition metrics. Indeed, I’m not sure how the particular re-weighting relates to repetition.

For PT – pairwise training – again while the particular synthetic data generation method proposed by the authors is nice, it seems from the naming and the experiments that the paper claims to be proposing the idea of a pairwise preference loss – e.g., it provides a new hinge-like likelihood-ratio loss, and the phrasing “pairwise training” is quite general. Yet, many many losses have been proposed for making use of a positive and negative answer pair (DPO, KTO, SimPO, etc., etc.,) and the choice of loss function seems independent from the synthetic data method the paper proposes (which itself depends on the paper’s definition of repetition.)

For the experiments, I don’t really understand the setting of taking a model, pruning it, finetuning it on Alpaca, and then evaluating it on wikitext PPL and generation and GLUE-style benchmarks (boolq, PIQA, Winogrande.) Unfortunately, my reading of the non-repetition metrics is that the methods don’t really help - -e.g., the changes in the various GLUE benchmarks are very small and often negative, and I’m not convinced in the importance of the repetition metrics in the setting provided. E.g., if you evaluate the model on Alpaca questions, does it degenerate into repetition? Surely not if you finetune it well?

Overall, the improvements of the methods seem to rest on the repetition metric introduced by the authors, and n-gram uniqueness, when generating wikitext. But if we care about the model generating good wikitext, why is the model itself not being finetuned on wikitext instead of Alpaca?

**Questions:**

Notes:
 - Please use \citep for citations that should be surrounded by parentheses. For example, “Curie et al. 2025 showed that” should be \citet, but “This has been observed in multiple studies (Curie et al., 2025, Newton et al., 2025)” should be \citep.
 - What is the “Wiki” dataset? – there are many Wikipedia-derived datasets. (e.g., line 053)
 - Top-k sampling was not introduced by Li et al., 2020 as cited. It was introduced by Fan et al., 2018 (Hierarchical Neural Story Generation.)
 - Shouldn’t CREP also depend on the n-gram chosen –  r ?
 - I’m confused about the Alpaca finetuning setup – you say you finetune on the Alpaca dataset, but then you have the model continue Wikitext-103 paragraphs. Is this in a question-answer format, like Alpaca?
 - What datasets are perplexity and MAUVE computed over?

---

> ### Author Response · Authors · 2025-11-20
> **Response to Reviewer 3Uto (Part 1)**
>
> Thank you for the detailed and valuable comments. We address comments here:
>
> ---
>
> > Please use \citep for citations that should be surrounded by parentheses. For example, “Curie et al. 2025 showed that” should be \citet, but “This has been observed in multiple studies (Curie et al., 2025, Newton et al., 2025)” should be \citep.
> > Top-k sampling was not introduced by Li et al., 2020 as cited. It was introduced by Fan et al., 2018 (Hierarchical Neural Story Generation.)
>
> Thank you for pointing out my mistakes. We have revised them accordingly in the updated PDF.
>
> ---
>
> > What is the “Wiki” dataset? – there are many Wikipedia-derived datasets. (e.g., line 053)
> > What datasets are perplexity and MAUVE computed over?
>
> As is conventional in other work, perplexity (PPL) is measured on the WikiText-2 dataset. When it comes to generation, 1,000 samples are randomly drawn from WikiText-103 and MAUVE is computed over the original sampled data and generated data. For clarity, we have revised the corresponding text in the paper.
>
> ---
> > Shouldn’t CREP also depend on the n-gram chosen – r ?
>
> We apologize for the confusion caused by the insufficient description of how CREP was measured in our experiments. In our evaluation, we used CREP in a more stringent manner to detect text degeneration. For each individual sentence, we computed CREP by exhaustively testing all r values from 4 to 16.
> We have now clarified this procedure in the experimental setup section and added the corresponding pseudocode in Appendix F.
>
> ---
> > For PT – pairwise training – again while the particular synthetic data generation method proposed by the authors is nice, it seems from the naming and the experiments that the paper claims to be proposing the idea of a pairwise preference loss – e.g., it provides a new hinge-like likelihood-ratio loss, and the phrasing “pairwise training” is quite general. Yet, many many losses have been proposed for making use of a positive and negative answer pair (DPO, KTO, SimPO, etc., etc.,) and the choice of loss function seems independent from the synthetic data method the paper proposes (which itself depends on the paper’s definition of repetition.)
>
> Thank you for pointing this out. We agree that the term pairwise training(PT) may have been overly general and could lead readers to interpret our method as proposing a new generic pairwise preference loss. To avoid misunderstanding, we will revise the method name to clearly reflect its purpose and scope, and we will update the text. We apologize for the confusion.

---

> ### Author Response · Authors · 2025-11-20
> **Response to Reviewer 3Uto (Part 2)**
>
> > I’m confused about the Alpaca finetuning setup – you say you finetune on the Alpaca dataset, but then you have the model continue Wikitext-103 paragraphs. Is this in a question-answer format, like Alpaca
>
> A common practice in pruning literature is to perform Alpaca-style finetuning after pruning and then evaluate the model’s general knowledge retention using WikiText perplexity. This is because maintaining broad linguistic and factual knowledge remains important even after the pruning–finetuning pipeline. In our case, we followed this convention for fairness to demonstrate that our repetition-mitigation methods do not introduce significant degradation, as reflected by the stable PPL on WikiText. [1-6]
>
> However, PPL alone is insufficient for assessing whether a model actually generates coherent and high-quality text. This is why we additionally evaluate WikiText-103 Continuation. Even if a model shows good perplexity, this does not necessarily guarantee strong generation quality within the same domain. As shown in our experiments, although our method leads to a slight increase in PPL, the distributional quality of the generated text, measured by MAUVE, consistently improves. For example, in the LLaMA-3.1-8B setting (Table 3), PPL increases from 21.69 to 23.12, while MAUVE improves from 0.61 to 0.73.
>
> The GLUE benchmark metrics are reported because they are commonly used in other pruning studies. Our goal in including them is to show that, even after applying additional training methods designed to reduce repetition, the resulting loss of general knowledge remains negligible. In other words, the performance with our method remains comparable to or not significantly worse than the performance without applying these repetition-mitigation techniques.
>
> Additionally, as the reviewer suggested, we examined whether repetition occurs when the model generates text using the same template employed during finetuning. To evaluate this, we applied the Alpaca template to the Self-Instruct dataset and generated answers. Then we measured the unique n-gram metric.
>
> | Method    | 3-gram | 4-gram | 5-gram | 6-gram |
> |-----------|--------|--------|--------|--------|
> | KD        | 11.91  | 9.35   | 7.75   | 6.68   |
> | | | | | |
> | FOCUS     | 6.59   | 4.26   | 3.07   | 2.36   |
> | +UL    | 5.42   | 3.32   | 2.29   | 1.71   |
> | +SG    | 5.47   | 3.50   | 2.50   | 1.96   |
> | +DITTO | 5.69   | 3.40   | 2.25   | 1.61   |
> | +PT  | 4.69   | 2.79   | 1.88   | 1.37   |
>
> In instruction-based generation, the input questions vary significantly in length, and very short responses can bias the evaluation because they contain too few n-grams. To avoid this issue, we filtered out responses shorter than 50 tokens. After applying this threshold, each method yielded roughly 500–600 valid samples, and we report the results in the Table below.
>
> | Method    | 3-gram | 4-gram | 5-gram | 6-gram |
> |-----------|--------|--------|--------|--------|
> | KD        | 21.14  | 16.98  | 14.33  | 12.53  |
> |           |        |        |        |        |
> | FOCUS     | 9.53   | 6.24   | 4.55   | 3.53   |
> | +UL       | 7.61   | 4.64   | 3.21   | 2.41   |
> | +SG       | 8.01   | 5.16   | 3.75   | 2.97   |
> | +DITTO    | 8.09   | 4.83   | 3.21   | 2.29   |
> | +PT       | 6.53   | 3.85   | 2.59   | 1.90   |
>
> As shown in the experimental results, using the same template as in finetuning still leads to a high rate of repetition (e.g., 3-gram: 21.14). When our method is applied, however, the repetition is reduced substantially, demonstrating its effectiveness even under template-matched generation.
>
> **
>
> [1]  Kim, B. K., Kim, G., Kim, T. H., Castells, T., Choi, S., Shin, J., & Song, H. K. (2024). Shortened llama: A simple depth pruning for large language models. arXiv preprint arXiv:2402.02834, 11, 1.
>
> [2] Ma, X., Fang, G., & Wang, X. (2023). Llm-pruner: On the structural pruning of large language models. Advances in neural information processing systems, 36, 21702-21720.
>
> [3] Kong, J., Ma, X., Wang, J., & Zhang, X. (2025, April). Sample-aware Adaptive Structured Pruning for Large Language Models. In Proceedings of the AAAI Conference on Artificial Intelligence (Vol. 39, No. 17, pp. 17938-17946).
>
> [4]  Gao, S., Lin, C. H., Hua, T., Tang, Z., Shen, Y., Jin, H., & Hsu, Y. C. (2024). Disp-llm: Dimension-independent structural pruning for large language models. Advances in Neural Information Processing Systems, 37, 72219-72244.
>
> [5] Xia, M., Gao, T., Zeng, Z., & Chen, D. (2023). Sheared llama: Accelerating language model pre-training via structured pruning. arXiv preprint arXiv:2310.06694. (ICLR 2024)

---

> ### Author Response · Authors · 2025-11-20
> **Response to Reviewer 3Uto (Part 3)**
>
> > For FOCUS for example, the hypotheses seems to be that KL-divergence regularization to a teacher doesn’t focus enough on tokens that are low-probability under the teacher (because the importance weight, the teacher’s probability, is close to 0.) KL-divergence does penalize putting probability mass on tokens that are low-probability under the teacher through the partition function of the softmax – the probabilities must sum to 1, so putting too-high probability on a low-probability token is penalized. The hypothesis here is unclear to me because we don’t see evidence in this paper that the particular weighting of KL is bad in this setting (the pruning-recovery setting) beyond FOCUS leading to higher scores under the authors’ proposed repetition metrics. Indeed, I’m not sure how the particular re-weighting relates to repetition.
>
> As the reviewer correctly pointed out, the mathematical form of KL ($\sum_x p_t(x)\,\log\left(\frac{p_t(x)}{p_s(x)}\right)$) shows that when the teacher assigns a very low probability to a token, the penalty incurred by the student assigning a high probability to that token becomes extremely small. This leads to insufficient learning in that region, and this property of KL is well known in the literature. Here, we simply note that the student may not sufficiently suppress tokens that the teacher intends to exclude. FOCUS provides a way to strengthen the learning signal in these regions when necessary.
>
> As discussed in Section 3.2, the capacity gap between the teacher and the pruned student makes it difficult for the student to learn all parts of the teacher distribution accurately. For this reason, we emphasize the high-confidence regions (i.e., probabilities close to 1 or 0), even if this leads to a slight increase in perplexity in the mid-probability range. By doing so, the student is more likely to retain valid candidates within the top-p or top-k region during sampling while avoiding tokens that should not be selected, including those that tend to trigger repetition.

---

### Official Review · Reviewer_MFvz · 2025-11-01

**Soundness:** 4
**Presentation:** 4
**Contribution:** 3
**Rating:** 8
**Confidence:** 5

**Summary:**

Addresses text degeneration, specifically repetition, which often worsens after pruning Large Language Models (LLMs).

Pruning reduces LLM size and latency but exacerbates repetition despite preserving perplexity that standard post-pruning fine-tuning fails to suppress.

The authors propose two novel token-level guidance methods:

1. FOCUS (Token-Weighted Distillation): This method uses token-weighted distillation to focus on high-confidence regions (where 0 and 1 are confident response probs and it decreases as we approach 0.5). It better aligns the smaller model with the larger teacher, reducing the likelihood of repetitive tokens.

2. PT (Pairwise Training): This method employs contrastive training with negative and positive samples. It explicitly encourages the model to generate alternative, more diverse tokens.

Experiments show these methods substantially reduce repetition and improve generation diversity in pruned LLMs.

The guidance methods minimally impact perplexity and enhance the performance of other training strategies.

**Strengths:**

- solves a critical problem (text degeneration / repetition) in pruned LLMs. Even though the discussion is about pruned LLMs, it seems to be widely applicable to any sort of distillation.
- Introduces two novel methods (FOCUS and PT) to approach the same.
- Experiments show the above methods to be very effective.
- Compatible with most existing training setups.

**Weaknesses:**

- Complexity of training increases due to the need of pairwise training data extraction.
- Adds multiple hyperparameters to tune.
- Encouraging the model to prefer the alternate tokens might confuse the model and lead to increased perplexity.
- Need to test how this impact factuality benchmarks where alternate tokens might be risky.

**Questions:**

see weakness section.

---

> ### Author Response · Authors · 2025-11-20
> **Response to  Reviewer MFvz**
>
> Thank you for the insightful comments. We address questions:
>
> ---
> > Complexity of training increases due to the need of pairwise training data extraction.
>
> As the reviewer noted, generating additional pairwise data incurs extra computational cost. Nevertheless, as shown in Figure 3(b) in the analysis section, the amount of data required is relatively small, approximately 1/13 of the full dataset (about 4K vs. 52K examples).
>
>
> ---
> > Encouraging the model to prefer the alternate tokens might confuse the model and lead to increased perplexity.
>
> > Need to test how this impact factuality benchmarks where alternate tokens might be risky.
>
> Encouraging the model to select alternative tokens can lead to a slight increase in perplexity(PPL) . We agree that distributional shifts may make the model’s behavior less stable and could weaken its adherence to factual patterns, thereby increasing the risk of hallucination. We evaluated this issue by running the Llama-3.1-8B models used in the paper on the TruthfulQA-MC2 benchmark.
>
> | Method | X     | UL    | SG    | DITTO | PT    |
> |--------|-------|-------|-------|--------|-------|
> | **KD**     | 0.384 | 0.389 | 0.384 | 0.385 | 0.398 |
> | **FOCUS**  | 0.412 | 0.425 | 0.417 | 0.417 | 0.434 |
> * X denotes that KD/FOCUS is only applied.
>
> According to the results, FOCUS achieved higher scores than standard KD in general. As mentioned in Section 3.2, this suggests that simply matching the teacher’s distribution may not be well aligned with the student’s limited capacity, which is consistent with prior work proposing modified or adaptive forms of KD. In addition, while perplexity is a common evaluation metric, a slight increase in PPL does not necessarily imply degraded generation quality. We will include a discussion of these additional experiments in the analysis section.

---

### Author Response · Authors · 2025-11-27
**Summary of Revisions**

We sincerely thank all reviewers for their thoughtful and constructive feedback.

With a few days remaining in the discussion period, we provide a concise summary of the major revisions and clarifications made throughout the manuscript. This summary is intended to help reviewers easily verify the updates applied in response to comments.

1. method name (PT)

As the reviewer pointed out, the original name Pairwise data Training (PT) could unintentionally give the impression that the paper proposes the general concept of pairwise training itself, which is indeed too broad. The core contribution of our method lies not in the pairwise formulation itself, but in the mechanism that detects when repetition begins and provides corrective signals by promoting more appropriate alternative tokens. To reflect this focus more accurately, we renamed the method to RePAIR (Repetition-aware PAIRwise alignment) and updated the manuscript accordingly. We apologize for any confusion caused by the earlier naming.

2. perplexity and performance

As several reviewers noted, applying our method leads to a slight increase in perplexity, raising concerns about potential side effects such as hallucination. To address this, we added an evaluation on the TruthfulQA benchmark in Section 6 of the revised manuscript. This experiment examines potential effects on factuality, and the results indicate that the method does not lead to increased hallucination or related issues.

3. DPO vs RePAIR with CE baseline

Several reviewers observed that RePAIR shares a similar pairwise training approach used in methods such as DPO. We clarified these similarities and differences in Section 6 of the revised manuscript and additionally included comparative experiments. For a fair comparison across methods, we also incorporated a CE baseline, highlighting the improvements provided by both DPO and RePAIR.

4. Additional experiments

As suggested by reviewers, we extended our analysis to verify whether the method remains effective under different pruning ratios. To this end, we conducted additional experiments with more aggressive pruning settings, and the results are included in Appendix G.

5. Detailed information and other revisions

- We added pseudocode and detailed descriptions of the experimental settings for the CREP metric in the appendix.
- We clarified the usage of “Wiki” in the main text to avoid ambiguity.
- We corrected several typos and minor issues pointed out by the reviewers.

---

### Comment · Area_Chair_9vbQ · 2025-11-28

Dear Reviewers,

The authors have responded to your reviews. Please engage in the discussion and evaluate the authors’ rebuttal to check whether your comments have been adequately addressed, and determine whether you would like to adjust your evaluations.

Best,

Your AC

---

### Meta-Review · Area_Chair_XtSy · 2026-01-06

**Summary:**

While compressing LLMs effectively, LLMs often result in text degeneration like repetition. To suppress this side effect, the paper introduces FOCUS and RePAIR, two token-level guidance methods that use distillation and contrastive training, respectively, to substantially reduce repetition and improve generation diversity. The reviewers have very mixed opinions, ranging from clear acceptance to clear rejection. The positive review is brief and somewhat general. While even the negative reviewer agrees that there are valuable contributions,this also points out a weak novelty. The reviewer also points out that using WikiText only might be weakness, ca ombined approach (PPL on WT-2, MAUVE on WT-103, and implicit use of Instruction-based generation tasks), however, is ok as it covers the foundational language model quality, the specific generation flaw (repetition/diversity) and instruction-following ability. Having said this, one could always provide better evaluations. The main concern for me was pointed out by two reviewers, namely that the loss and the pairwise training proposed are not really novel. I do agree that this needs more clarification, though it is interesting to see that this helps in general. However, there is a related work that should be discussed in more detail. Nevertheless, this is interesting work that seems to work.

**Reviewer Concerns:**

The main concern raised is low novelty. This has not been fully addressed by the rebuttal. The authors provided additional evaluations, which addresses concerns about evaluation comprehensiveness to some extent. However, they did not sufficiently position their work within the global research landscape or clarify what is truly novel about their approach compared to existing methods. The novelty concerns, specifically regarding the loss function and pairwise training, are not addressed. However, it is interesting to observe that these methods help in practice. The question of what distinguishes this work from related approaches needs more detailed discussion and clarification.

**Reviewer Scores:**

I do not expect significant score changes. The positive reviewers, who provided a brief and general assessment, would likely maintain their acceptance score, as the additional evaluations support their positive view. The negative reviewer, who raised concerns about novelty, would likely not change the score, as the core novelty issues were not adequately addressed (the authors did not clarify what makes their approach distinct from existing methods or provide the necessary related work discussion). The reviewer in the middle range might remain unchanged or shift slightly, but without addressing the novelty concerns and better positioning within the research landscape, no substantial score improvements are expected.

---

### Decision · Program_Chairs · 2026-01-26

Reject